# Unabsorbed Slack Resources and Enterprise Innovation: The Moderating Effect of Environmental Uncertainty and Managerial Ability

Yan Zhang, Ziyuan Sun *  and Mengxin Sun

School of Economics and Management, China University of Mining and Technology, Xuzhou 221116, China; tb20070017b4@cumt.edu.cn (Y.Z.); ts21070285p31tm@cumt.edu.cn (M.S.)
* Correspondence: zyscumt2021@126.com

**Abstract:** Unabsorbed slack resources are critical for organizational innovation, but research concerning the relationship between unabsorbed slack and corporate innovation has resulted in controversial findings. Using the data of all A-shared listed companies in China from 2011 to 2018, this paper examines the influence of unabsorbed slack resources on enterprise innovation. First, the paper verifies that there is an inverted U-shaped relationship between unabsorbed slack resources and R&D investment and innovation output. Following that, empirical findings show that environmental uncertainty and managerial ability have a negative incentive effect on the relationship between unabsorbed slack resources and enterprise innovation. Finally, the results of mechanism testing reveal that unabsorbed slack resources affect the enterprise innovation through two channels: resource effect and agency cost.

**Keywords:** unabsorbed slack resources; enterprise innovation; moderating effect; managerial ability; environmental uncertainty





## 1. Introduction

Technological innovation plays an important role in the success of enterprises [1] and it is a critical factor for them to gain strong short-term market performance and long-term competitive advantage [2]. Due to long cycles, large investments [3], and high adjustment costs, adequate resource support is essential to ensure the sustainability of innovation activities. However, in the Chinese capital market, the inadequate financial system and information asymmetry problems make it difficult for most enterprises to obtain innovation resources from external sources. Consequently, the role of internal resources, i.e., slack resources, is becoming more and more important for innovation.

In 1963, Cyert & March defined slack resources as "the difference between total resources needed by the enterprise organization to maintain the status quo and the resources actually possessed by the organization" [4]. There exist different forms of slack resources, such as idle machinery and equipment, surplus cash, extra employees, and semi-finished products in processing. According to liquidity and flexibility, the slack resources are divided into unabsorbed slack resources and absorbed slack resources [5]. Unabsorbed slack resources with strong liquidity are not absorbed within the enterprise, and they can be used to cope with market competition and institutional pressure in a competitive environment, meet diversified resource demands, and help enterprises respond rapidly to environmental changes [6]. In contrast, absorbed slack resources have been internalized in the enterprise process with poor liquidity and strong specificity. They can only be used for specific purposes and are not easily reconfigured, making it difficult for managers to convert them into specific resources required by innovative activities in a short period of time [7].

Scholars have performed extensive research on how slack resources influence enterprises' technological innovation. According to Wei et al. (2020), slack resources in enterprises can be seized by managers as an informal way to provide resource support for enterprises' innovation activities [8]. Through making flexible use of slack resources, enterprises can attempt innovation strategies so as to promote technological innovation. However, other scholars confirm that although some enterprises possess some slack resources, their technological innovations are still lacking resource support [9]. The above controversy stems from the insufficient consideration of the characteristics of different slack resources; the existing literature generally treats them as a whole without distinguishing between them [10]. Therefore, it is urgent to explore the relationship between different slack resources and corporate technological innovation. Moreover, in the modern business world, an enterprise will not survive without the external environment. Does the relationship between slack resources and corporate technological innovation change as the uncertainty of the external environment increases? At the same time, managers are the decision-makers of enterprise innovation activities, and their competencies play a critical role in the success of technological innovation. Do different managerial competencies influence the relationship between slack resources and enterprise technological innovation? Obviously, the existing literature does not provide answers to these questions.

To address the research gaps, this paper divides slack resources into absorbed and unabsorbed slack resources and explores the relationship between unabsorbed slack resources, which are most closely related to innovation and can be flexibly used by enterprises and enterprise technological innovation. This paper attempts to focus on the following questions: Firstly, how do unabsorbed slack resources influence enterprise technological innovation, and what are the mechanisms? Secondly, do external environmental uncertainty and internal managerial ability affect the relationship between unabsorbed slack resources and enterprise technological innovation?

The main contributions of this paper are threefold. Firstly, this paper makes a classification of slack resources and explores the influence of unabsorbed slack resources on enterprise innovation, addressing the dispute resulting from general treatment without differentiation of slack resources [11]. Secondly, the mechanisms of the inverted U-shaped impact of unabsorbed slack resources on R&D investment and innovation output are examined from the perspective of resource effect and agency cost, which enhances the theoretical basis of the smoothing effect of unabsorbed slack resources on enterprise innovation. Thirdly, in terms of environmental uncertainty and managerial ability, this paper reveals the moderating role of internal and external factors on the relationship between unabsorbed slack resources and enterprise innovation, reveals the mechanisms by which enterprises choose to accumulate or consume unabsorbed slack resources in different contexts, and broadens the boundaries of the existing literature on the factors influencing technological innovation.

The structure of the remainder of the paper is as follows. The second part is the literature review and research hypothesis. The third part is the research design, including sample selection, data sources, variable definitions, and empirical methods. The fourth part is empirical research results and analysis, including descriptive statistics of main variables, correlation analysis, regression analysis, and robustness tests. The fifth part is the mechanism tests. Finally, the research conclusions and policy implications are given.

## 2. Literature Review and Research Hypothesis

### 2.1. Unabsorbed Slack Resources and Enterprise Innovation

Innovation requires a major expenditure of resources, and resource availability is the critical factor of innovation success [12]. The more resources an enterprise owns, the more autonomous it will be when pursuing market opportunities and the more likely it will be to make disruptive innovations [13]. However, R&D investment is different from the ordinary [14], with the characteristics of a large amount, a long cycle, and uncertain results [15]. Enterprises engaged in innovation are prone to the problem of insufficient

funding for internal R&D [16]. Given the high adjustment costs of innovation activities [17], enterprises generally have a strong incentive to maintain a continuous level of innovation investment (innovation smoothing), which requires continuous and adequate financial support. Due to the positive externalities of innovation activities, innovative enterprises are usually unwilling to disclose information concerning innovation [18]. This makes it difficult for external capital providers to properly evaluate the real value of innovation projects, so they are naturally reluctant to lend funds to enterprises, leading to innovative enterprises facing serious financing constraints.

In this context, from the perspective of organization theory, as the abundant security resources of the enterprise [19], unabsorbed slack resources can provide important support for enterprises to take risks, make positive strategies and maintain a competitive edge. They can play a buffer role when enterprises face financing constraints: they provide relatively stable cash flow for enterprises' innovation activities, which can be effectively converted to maintain and restore enterprise productivity and help enterprises cope with the adverse impact from external environmental uncertainty and alleviate financing constraints in a timely manner. Thus, innovation activities can be carried out continuously [20], and the sustainability of R&D investment and innovation output of innovation activities can be maintained. On the other hand, unabsorbed slack resources are of high liquidity, which means lower adjustment costs. When enterprises face financing constraints, they can increase financial flexibility by appropriately reducing unabsorbed slack resources to seize opportunities from environmental changes. This helps organizations to innovate in uncertain external situations, try new strategic changes and innovation activities, keep enterprise innovation ability [4], and build innovation atmosphere [21]. Finally, the fluctuation of investment in enterprise innovation is relatively flat compared with the negative shock of cash flow. Therefore, unabsorbed slack resources increase innovation resources to a certain extent and smooth part of the fluctuation of innovation activities due to financing constraints.

However, an enterprise has an expectation of the number of slack resources [22], which may be affected by industry environment, internal operation efficiency, and other aspects of the enterprise. From the perspective of agency theory, when the actual number of slack resources is greater than expected, enterprises tend to search and seize external opportunities to consume these resources, inducing "opportunity searching" behavior to restore the number of slack resources to the expected level. This often leads to the phenomenon that slack resources spawn the irrational use of resources, considered by the agency theory [23], such as the expansion of management rights and excessive investment. They even lead to the shareholder who holds most of the enterprise's equity to transfer the enterprise's property and other resources for their own interests, namely the large shareholder tunneling. The large shareholder tunneling directly damages the interests of minority shareholders and the enterprise's future growth opportunities, which is not conducive to the enterprise's future investment in innovation activities. In addition, excessive slack resources are synonymous with "low efficiency" of enterprises [24], indicating that enterprises lack initiative in trying innovative breakthroughs. As a result, when there are too many unabsorbed slack resources, the enterprise's innovative activities suffer.

This leads to the following hypotheses of this paper:

**H1:** *There is an inverse U-shaped relationship between unabsorbed slack resources and R&D investment.*

**H2:** *There is an inverse U-shaped relationship between unabsorbed slack resources and innovation output.*

### 2.2. Unabsorbed Slack Resources, Environmental Uncertainty, and Enterprise Innovation

Environmental uncertainty is the complexity of changes in an organization's external environment and stands for the organization's responsiveness [25]. Specifically, environmental uncertainty is measured by the degree of change in the external environment in terms of "technology, customer preference, and product demand or material supply" [26].

A complex external environment will put huge pressure on the enterprise. The higher the uncertainty of the environment, the faster the change of the external environment, and the higher management risk that the enterprise faces, the more urgent it becomes for enterprises to respond quickly to changing circumstances [27].

In modern commercial society, an enterprise cannot survive without the external environment, and environmental factors need to be taken into account while making decisions. When the external environment is unpredictable, enterprises face increased operational risks [28], which makes their decision-making more cautious. As a consequence, enterprises must manage the scarce resources they possess. Unabsorbed slack resources are used to maintain and restore the productivity of enterprises. Moreover, they are transformed into marketing and other business and management activities, even into absorbed slack resources. Because absorbed slack resources are internalized in the enterprises' business management activities, they are not flexible and have the characteristics of "low efficiency or even waste" in the view of agency theory, leading directly to the gap between the actual output and the maximum output of the enterprise innovation activities.

Additionally, in the dynamic environment of environmental uncertainty, the information obtained from the outside is characterized by limited, lagging, and lower accuracy. This characteristic makes managers face decision fuzziness and affects their ability to judge the future [29]. According to social learning theory, this high level of environmental uncertainty can trigger a "herd effect," whereby management tends to adopt imitation strategies to reduce costs [30] and is more likely to adopt conservative strategies to maintain cash flow and reduce R&D investment. Therefore, when other factors are equal, the higher the degree of environmental uncertainty, the fewer unabsorbed slack resources will be applied to the maintenance of enterprise innovation, leading to a reduction in innovation input and output.

Based on this, this paper puts forward the following hypothesis:

**H3:** *Environmental uncertainty negatively moderates the relationship between unabsorbed slack resources and enterprise innovation. That is, the higher the environmental uncertainty, the weaker the incentive effect of unabsorbed slack resources on enterprise innovation.*

### 2.3. Unabsorbed Slack Resources, Managerial Ability, and Enterprise Innovation

Managers are some of the most important stakeholders in modern enterprises. The ability of managers to generate revenue under resource constraints is called "managerial ability" [31]. According to upper echelon theory, the ability of management, as an important human capital, is a critical factor for innovation success. However, at present, the relationship between managerial ability and enterprise innovation is equivocal [32].

As an effective way for enterprises to obtain and maintain their competitive advantages, innovation has the characteristics of high risk, long cycle, and high investment, making it an uncertain decision for management [33]. There are two opposing views on how managerial ability influences enterprise innovation [32]. Some scholars argue that managerial ability has a positive impact on enterprise innovation. From the risk tolerance perspective, strong managers are competent in risk control, resource integration, opportunity discovery, and learning skills [34] and are more likely to invest in high-risk projects, that is, a certain degree of risk taking. A high level of risk-taking contributes to improving innovation performance and to pursuing high-risk innovation projects [35]. Simultaneously, competent managers are more experienced in managing the enterprise's resources. Through strong ability of resource integration and allocation, enterprises can improve risk-tolerance, leverage unabsorbed slack resources, and better execute innovation projects [36]. In addition, more competent management can stimulate researchers to realize their full creative potential, thereby providing the most valuable human resources for innovation and improving innovation performance.

Contrarily, other scholars hold that managerial ability has a negative impact on innovation [37]. Based on the principal-agent theory, due to information asymmetry, the manager tends to avoid risk in investment decision-making out of personal self-interest

rather than "organizational benefits maximization." The more competent a manager is, the more likely he or she will benefit from avoiding venture capital [38]. The agency problem causes slack resources to be regarded as a tool for managers to achieve their personal goals, and managers are prone to resource satisfaction, which in turn weakens the incentive to innovate and reduces the R&D investment [39]. Additionally, based on the management defense theory and reputation theory, the absence of constraints in the internal and external governance structure makes managers with strong competencies focus more on their reputation and future career development; therefore, they will avoid high-risk projects, take a more conservative approach to resource use, and reduce innovation [37]. Additionally, the way in which enterprises reduce agency costs by executive stockholding brings a certain convergence effect and further leads to a management defense effect [40]. The stronger the managerial ability one has, the more serious the defense effect is; when the equity incentive exceeds a certain range, the higher power and managerial ability means the higher residual income claim, and this will have a much stronger "tunneling" motivation for a large number of unabsorbed slack resources [41] and reduce investment in long-term projects, such as enterprise innovation.

Based on the above analysis, according to the "Risk preference" and "Risk aversion" viewpoints, this paper puts forward the following competitive hypotheses:

**H4:** *Managerial ability positively moderates the relationship between unabsorbed slack resources and enterprise innovation. That is, the stronger the managerial ability, the higher the risk-tolerance level of the enterprise and the stronger the incentive effect of unabsorbed slack resources on enterprise innovation.*

**H5:** *Managerial ability negatively moderates the relationship between unabsorbed slack resources and enterprise innovation. That is, the stronger the managerial ability, the lower the risk-tolerance level of the enterprise and the weaker the incentive effect of unabsorbed slack resources on enterprise innovation.*

Figure 1 illustrates the research framework of this study.

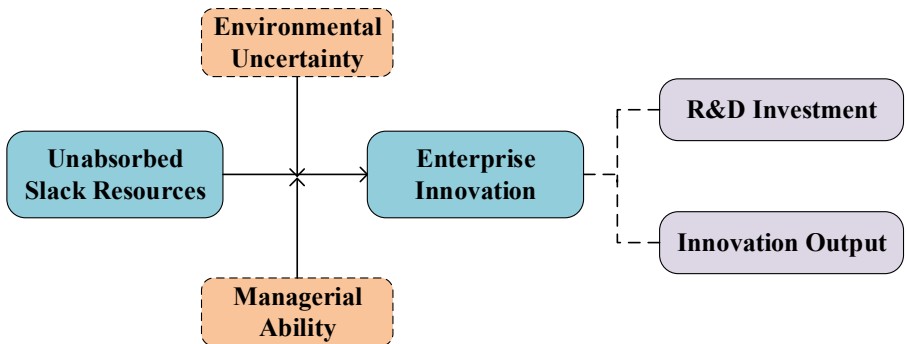

**Figure 1.** Research framework.

## 3. Research Design

### 3.1. Sample Selection and Data Sources

We selected all A-share listed companies in Shanghai and Shenzhen Stock Exchanges from 2011 to 2018 as research samples. The 2011 starting year is predicated upon data validity, with corporations generally providing few innovation inputs and patent data for the pre-2011 period. The samples were screened according to the following principles: (1) remove the samples with financial leverage (Lev) less than 0 or greater than 1, (2) remove financial listed companies, (3) remove samples with missing data, (4) remove *ST* and *PT* companies. In order to eliminate the influence of extreme values during actual regression, this paper winsorizes all continuous variables according to the standard of 1%. The financial data involved in this paper are mainly from the China Stock Market & Accounting Research

Database. Some missing data were gathered by manually consulting the annual report, and 8236 observations in 8 years were finally obtained.

*3.2. Variable Definitions*

3.2.1. Explanatory Variable

Compared with absorbed slack resources, unabsorbed slack resources have higher liquidity and flexibility in operation. Therefore, this paper adopts quick ratio to measure unabsorbed slack resources [42], and the specific calculation formula is (current assets—inventory)/current liabilities.

3.2.2. Explained Variables

This paper measures enterprise innovation through R&D investment and innovation output. R&D investment (Rd_assets) is measured by dividing the R&D investment amount of the current year by the total ending assets. Innovation output adopts the method of adding 1 to the number of invention patents obtained in a year and then taking the natural logarithm.

3.2.3. Moderating Variables

Environmental uncertainty (EU) is an indicator of environmental uncertainty. The root cause of environmental uncertainty lies in the external environment, and the change in the external environment will cause a fluctuation in the core business activities of the enterprise and ultimately lead to a fluctuation in sales revenue. Therefore, environmental uncertainty can be measured by the fluctuation of the enterprise's performance. In order to eliminate the influence of the industry, the standard deviation of sales revenue is generally considered to be a measure of environmental uncertainty. Ghosh & Olsen (2009) and Panousi & Papanikolaou (2012) used the standard deviation of sales revenue over the past five years with industry adjustment to measure the environmental uncertainty of the enterprise [43,44]. However, part of the changes in sales revenue over the past 5 years were brought by enterprises' steady growth. Accordingly, in order to measure the environmental uncertainty more accurately, it is necessary to exclude the stable growth part of sales revenue, that is, each enterprise uses the data of the past 5 years and adopts ordinary least squares (OLS) to run the following Model (1) to estimate the abnormal sales revenue of the past 5:

$$Sales_{i,t} = \alpha_0 + \alpha_1 Year + \varepsilon_{i,t} \tag{1}$$

Here, Sales refers to sales revenue. Year is an annual variable. If the observed value is from the last 4 years, then Year = 1. If the observed value occurred within the last 3 years, then Year = 2; similarly, if the observed value occurred within the current year, then Year = 5. The residual of Model (1) is abnormal sales revenue. This paper calculates the standard deviation of abnormal sales revenue of the enterprise in the past 5 years and divides it by the average sales revenue of the past 5 years to obtain the environmental uncertainty without industry adjustment. The median of the unadjusted environmental uncertainty of all enterprises in the same industry in the same year is the industry's environmental uncertainty. Then, dividing the unadjusted environmental uncertainty of each enterprise by the industry's environmental uncertainty gives the environmental uncertainty adjusted by the industry.

Referring to the practice of Demerjian [31], DEA method combined with Tobit regression is adopted to measure managerial ability. The specific calculation steps are as follows. First, the net fixed assets, net intangible assets, net goodwill, R&D expenditure, operating cost, and the sum of sales expense and administrative expense are taken as the input variables in the DEA analysis, and the operating income is taken as the only output variable. The enterprise efficiency value is calculated by the DEA_Solverpro5.0 software. Then, the Tobit model is used to separate the enterprise efficiency value from both the enterprise level and the management level. The residual of regression is MA_Socre. Among them, the enterprise-level factors controlled in the Tobit model include enterprise

size, market share, free cash flow, age of establishment, degree of internationalization, and degree of diversification. In addition, in the robustness test, the regression residuals are divided from small to large into four groups, with values of 1, 2, 3, and 4, respectively, recorded as MA_Rank, to re-measure managerial ability.

Variable definitions are shown in Table A1.

### 3.3. Empirical Methods

In order to test the impact of unabsorbed slack resources on enterprise innovation, following Jin et al. (2021) [45], we establish Model 2:

$$Rd\_assets_{i,t}/Patent_{i,t} = \beta_0 + \beta_1 Slack_{i,t} + \beta_2 Slack^2_{i,t} + \sum Control_{i,t} + \mu_{i,t} \tag{2}$$

Here, $i$ indexes the enterprise, $t$ indexes year, $\sum Control_{i,t}$ indexes the series of control variables mentioned above, and $\mu_{i,t}$ is the random error term. $Slack^2$ represents the square term of unabsorbed slack resources, which is used to describe its non-linear impact on enterprise innovation. In Model (2), this paper mainly focuses on the coefficient of $\beta_2$; if $\beta_2$ is significant, it means that the non-linear relationship assumed above is true and vice versa.

To verify the moderating effect of external environmental uncertainty and managerial ability on unabsorbed slack resources and enterprise innovation, following Yang (2021) [46], this paper constructs Model (3) and Model (4).

$$Rd\_assets_{i,t}/Patent_{i,t} = \gamma_0 + \gamma_1 Slack_{i,t} + \gamma_2 Slack^2_{i,t} + \gamma_3 Slack_{i,t} \times EU_{i,t} + \sum Control_{i,t} + \mu_{i,t} \tag{3}$$

$$Rd\_assets_{i,t}/Patent_{i,t} = \eta_0 + \beta_1 Slack_{i,t} + \eta_2 Slack^2_{i,t} + \eta_3 Slack_{i,t} \times MA\_Score_{i,t} + \sum Control_{i,t} + \mu_{i,t} \tag{4}$$

In Model (3) and Model (4), we focus on the intersection term coefficients $\gamma_3$ and $\eta_3$ of unabsorbed slack resources with environmental uncertainty and managerial ability, respectively. If the coefficient of $\gamma_3$ and $\eta_3$ is significantly greater than 0, it will indicate that the relationship between unabsorbed slack resources and enterprise innovation is positively moderated by environmental uncertainty or managerial ability, and if the coefficient is negative, it will negatively moderate. If $\gamma_3$ and $\eta_3$ are not significant, there will be no regulatory effect. To ensure the robustness of the results, industry dummy variables and time dummy variables are also added to the model.

## 4. Empirical Research Results and Analysis

### 4.1. Descriptive Statistics

Table 1 shows the descriptive statistical results of the main variables. The average value of enterprise R&D investment is 0.0209, indicating that the average R&D investment amount of the sample enterprises accounts for 2.09% of the total assets, and the overall R&D investment intensity is not high. Moreover, the minimum and maximum values of R&D investment are 0.0001 and 0.0927, respectively, indicating that there are great differences in the R&D investment intensity among different enterprises. In terms of innovation output, the mean value and median value are 1.4424 and 1.2501, respectively. The mean value is larger than the standard deviation, and the sample is right-skewed. The minimum value of innovation output is 0, and the maximum value is 5.1120. There is also a large gap in innovation output among different enterprises. The mean value of unabsorbed slack resources is 1.7088, and the median value is 1.17336, indicating that the absolute amount of unabsorbed slack resources of more than 50% of enterprises exceeds their current liabilities. The mean value of environmental uncertainty is 1.2500, and the standard deviation is 1.0307; the standard deviation is small. The minimum value of managerial ability is −0.3132, the median value is −0.0432 and the maximum value is 0.3731, indicating that there are great

differences in managerial ability among different enterprises. The values and distributions of other variables are basically consistent with the existing studies, indicating that the sample selection in this paper is reasonable.

**Table 1.** Descriptive statistics of the main variables.

| Variables | N | Mean | Sd | P25 | P50 | P75 | Min | Max |
|---|---|---|---|---|---|---|---|---|
| Rd_assets | 8236 | 0.0209 | 0.0174 | 0.0086 | 0.0177 | 0.0281 | 0.0001 | 0.0927 |
| Patent | 8236 | 1.4424 | 1.2501 | 0.0000 | 1.3863 | 2.1972 | 0.0000 | 5.1120 |
| Slack | 8236 | 1.7088 | 1.7336 | 0.7706 | 1.1806 | 1.9155 | 0.2214 | 12.3895 |
| EU | 8236 | 1.2500 | 1.0307 | 0.5871 | 0.9699 | 1.5356 | 0.1330 | 5.9394 |
| MA_Score | 8236 | −0.0267 | 0.1419 | −0.1261 | −0.0432 | 0.0557 | −0.3132 | 0.3731 |
| Size | 8236 | 22.3573 | 1.1924 | 21.5197 | 22.2044 | 23.0289 | 20.0580 | 26.0239 |
| Lev | 8236 | 0.4298 | 0.1958 | 0.2765 | 0.4234 | 0.5757 | 0.0552 | 0.8977 |
| Roa | 8236 | 0.0363 | 0.0610 | 0.0119 | 0.0334 | 0.0646 | −0.2315 | 0.2087 |
| Age | 8236 | 2.3674 | 0.5057 | 1.9459 | 2.3026 | 2.8332 | 1.0986 | 3.2189 |
| Board | 8236 | 2.1376 | 0.1947 | 1.9459 | 2.1972 | 2.1972 | 1.6094 | 2.7081 |
| Bm | 8236 | 0.5817 | 0.2461 | 0.3860 | 0.5678 | 0.7669 | 0.1197 | 1.1195 |
| Top1 | 8236 | 0.3298 | 0.1397 | 0.2217 | 0.3065 | 0.4214 | 0.0908 | 0.7210 |
| Grow | 8236 | 0.1712 | 0.3823 | −0.0190 | 0.1092 | 0.2648 | −0.4782 | 2.3539 |

*4.2. Correlation Analysis*

*Pearson* correlation coefficients of the main variables are shown in Table A2. It can be seen that the correlation coefficient between unabsorbed slack resources (Slack) and enterprise R&D investment (Rd_assets) is 0.161, and it is significant at the 1% level, preliminarily indicating that unabsorbed slack resources promote enterprise R&D investment. Unabsorbed slack resources (Slack) are significantly negatively correlated with innovation output (Patent), and the corresponding correlation coefficient is −0.095. It is significant at the 1% level, preliminarily indicating that unabsorbed slack resources inhibit innovation output. Meanwhile, this paper calculates the variance inflation factors of the main variables, the values of which are less than 3, with an average value of 1.56, confirming that there is no serious multicollinearity problem in this paper.

*4.3. Regression Analysis*

4.3.1. Unabsorbed Slack Resources and Enterprise Innovation

In order to test the impact of unabsorbed slack resources on enterprise innovation, this paper firstly makes a regression to Model (2). Results are reported in columns (1)–(4) of Table 2. Columns (1) and (2) are the impact of unabsorbed slack resources on R&D investment. It can be seen that regardless of whether control variables are added, the coefficient of $Slack^2$ is significantly negative, at least at the 5% level. Consistent with the previous studies, the results indicate that organization theory is more insightful when unabsorbed slack resources are in a relatively reasonable range, and agency theory is more significantly supported when unabsorbed slack resources are at a high level. Thus, the hypothesis H1 is established.

**Table 2.** Unabsorbed slack resources and enterprise innovation.

| Variables | (1) | (2) | (3) | (4) | (5) | (6) | (7) | (8) |
|---|---|---|---|---|---|---|---|---|
| | Rd_Assets | Rd_Assets | Patent | Patent | F.Rd_Assets | F.Rd_Assets | F.Rd_Assets | F.Patent |
| Slack | 0.0053 *** | 0.0012 ** | −0.0458 ** | 0.0780 ** | 0.0052 *** | 0.0010 | −0.0300 | 0.0927 ** |
| | (19.26) | (2.13) | (−2.29) | (2.13) | (16.47) | (1.60) | (−1.29) | (2.23) |
| Slack$^2$ | −0.0004 *** | −0.0001 *** | −0.0029 | −0.0100 *** | −0.0004 *** | −0.0001 * | −0.0044 * | −0.0116 *** |
| | (−14.46) | (−2.74) | (−1.43) | (−3.62) | (−12.17) | (−1.95) | (−1.91) | (−3.68) |
| Size | | 0.0001 | | 0.5570 *** | | −0.0000 | | 0.5798 *** |
| | | (0.22) | | (17.33) | | (−0.06) | | (15.87) |
| Lev | | 0.0021 | | −0.2651 | | 0.0026 | | −0.1880 |
| | | (0.80) | | (−1.41) | | (0.89) | | (−0.88) |
| Roa | | 0.0364 *** | | −0.2420 | | 0.0480 *** | | 0.3416 |
| | | (6.85) | | (−0.81) | | (6.69) | | (0.81) |
| Age | | −0.0015 ** | | −0.0420 | | −0.0012 | | −0.0500 |
| | | (−2.13) | | (−0.88) | | (−1.61) | | (−0.94) |
| Board | | −0.0008 | | 0.2585 ** | | −0.0001 | | 0.2842 ** |
| | | (−0.43) | | (2.16) | | (−0.04) | | (2.15) |
| Bm | | −0.0162 *** | | −0.7885 *** | | −0.0133 *** | | −0.8897 *** |
| | | (−7.52) | | (−6.20) | | (−5.44) | | (−5.83) |
| Top1 | | −0.0028 | | −0.0841 | | −0.0030 | | −0.0734 |
| | | (−1.13) | | (−0.47) | | (−1.09) | | (−0.37) |
| Grow | | −0.0013 *** | | −0.0255 | | −0.0007 | | 0.0263 |
| | | (−2.73) | | (−0.72) | | (−1.26) | | (0.65) |
| _cons | 0.0142 *** | 0.0173 * | 1.5375 *** | −11.9767 *** | 0.0145 *** | 0.0191 * | 1.6055 *** | −12.3838 *** |
| | (37.84) | (1.88) | (55.97) | (-18.59) | (33.18) | (1.87) | (49.57) | (−17.18) |
| Ind/Year | YES | YES | YES | YES | YES | YES | YES | YES |
| N | 8236 | 8236 | 8236 | 8236 | 6094 | 6094 | 6094 | 6094 |
| adj. R$^2$ | 0.050 | 0.280 | 0.010 | 0.282 | 0.051 | 0.280 | 0.010 | 0.287 |

Notes: T-statistics in parentheses are on the basis of standard errors clustered by firms and robust to heteroscedasticity. The *, **, and ***, respectively, denote the significance on the basis of two-tailed *t*-tests at or below the 10%, 5%, and 1% levels. The same below.

In addition, columns (3) and (4) show the impact of unabsorbed slack resources on innovation output. After adding control variables, the coefficient of Slack$^2$ is −0.0100, and it is significant at the 1% level, indicating that there is also an inverted U-shaped relationship between unabsorbed slack resources and enterprise innovation output. Therefore, H2 holds. In order to alleviate the endogenous problems caused by possible reverse causality, this paper further investigates the influence of independent variables on R&D investment and innovation output in the (T+1) period. The coefficient of Slack$^2$ is still significantly negative.

Lind and Mehlum (2010) point out that many U-shaped or inverted U-shaped relationships are merely theoretically true due to the actual value range of the explanatory variable being only on one side of the U-shaped or inverted U-shaped extreme point. In other words, real economic problems simply cannot reach the extreme point where the relationship between the explanatory variable and the explained variable reverses [47]. In order to solve this problem, we calculate the extreme point of the inverted U-shaped relationship according to the regression results of Table 2, and the results are shown in Table 3. The calculated extreme point is around the two-fifths number of unabsorbed slack resources (Slack), and there is no case that the extreme point cannot reach. In summary, unabsorbed slack resources first increase and then decrease with R&D investment and innovation output.

**Table 3.** The extreme point of unabsorbed slack resources on enterprise innovation.

| Enterprise Innovation | The Extreme Point of Unabsorbed Slack Resources (Slack) |
|---|---|
| Rd_assets | 4.9431 |
| F.Rd_assets | 5.3055 |
| Patent | 3.9013 |
| F.Patent | 3.9859 |

4.3.2. Moderating Effect Test of Environmental Uncertainty

To verify whether environmental uncertainty moderates the relationship between unabsorbed slack resources and enterprise innovation, we ran Model (3), and the results are shown in Table 4. When regressing R&D investment, the coefficient between the environmental uncertainty intersection term (Slack × EU) and the unabsorbed slack resources is significantly negative at the 1% level, whether current or next period, indicating that environmental uncertainty negatively moderates the inverted U-shaped relationship between unabsorbed slack resources and R&D investment. In terms of innovation output, the coefficient between intersection term (Slack × EU) and the current innovation output is $-0.01849$ ($p = -2.81$), and the coefficient between intersection term (Slack × EU) and the next innovation output is $-0.0228$ ($p = -3.58$). This finding is in line with organizational learning theory. High environmental uncertainty can stimulate enterprises to adopt imitation strategies and thus reduce innovation activities [30]. Specifically, the higher the environmental uncertainty, the weaker the incentive effect of unabsorbed slack resources on R&D investment and innovation output, which is consistent with the hypothesis H3.

4.3.3. Moderating Effect Test of Managerial Ability

Table 5 reports the impact of unabsorbed slack resources on enterprise innovation in the current and next period under different levels of managerial ability. As can be seen from columns (1) and (2), the coefficient of $Slack^2$ does not change after adding a moderating effect, indicating that the inverted U-shaped relationship between unabsorbed slack resources and enterprise innovation does not change with different managerial abilities. Simultaneously, the intersection term between unabsorbed slack resources and managerial ability (Slack × MA_Score) is significantly negatively correlated with both R&D investment and innovation output at the level of 1%. Similar to what was reported by Eisenmann (2010) and Hirshleifer et al. (2012), information asymmetry makes it easier for management to benefit from risk avoidance investment [37,48], so the more powerful the management is, the less willing it is to engage in innovative activities. In other words, the unabsorbed slack resources of the enterprise with stronger managerial ability have a weaker incentive effect on R&D investment and innovation output in the current period when other conditions remain unchanged. Furthermore, we make a re-regression of R&D investment and innovation output with a lag of one stage, as shown in columns (3) and (4), and find that the coefficient of intersection term (Slack × MA_Score) is still significantly negative at 1%. Therefore, H5 is supported.

**Table 4.** Unabsorbed slack resources and enterprise innovation: the moderating effect of environmental uncertainty.

| Variables | (1) Rd_Assets | (2) Patent | (3) F.Rd_Assets | (4) F.Patent |
|---|---|---|---|---|
| Slack | 0.0020 *** | 0.0981 *** | 0.0019 *** | 0.1166 *** |
| | (3.20) | (2.60) | (2.69) | (2.75) |
| Slack$^2$ | −0.0001 ** | −0.0096 *** | −0.0001 | −0.0110 *** |
| | (−2.37) | (−3.45) | (−1.56) | (−3.48) |
| Slack × EU | −0.0007 *** | −0.0184 *** | −0.0008 *** | −0.0228 *** |
| | (−5.52) | (−2.81) | (−6.20) | (−3.58) |
| Size | 0.0001 | 0.5573 *** | 0.0000 | 0.5810 *** |
| | (0.25) | (17.40) | (0.02) | (15.96) |
| Lev | 0.0016 | −0.2754 | 0.0020 | −0.2043 |
| | (0.65) | (−1.46) | (0.70) | (−0.95) |
| Roa | 0.0320 *** | −0.3508 | 0.0432 *** | 0.2044 |
| | (6.13) | (−1.17) | (6.16) | (0.48) |
| Age | −0.0012 * | −0.0356 | −0.0010 | −0.0427 |
| | (−1.77) | (−0.74) | (−1.29) | (−0.80) |
| Board | −0.0009 | 0.2544 ** | −0.0003 | 0.2772 ** |
| | (−0.52) | (2.13) | (−0.16) | (2.10) |
| Bm | −0.0163 *** | −0.7895 *** | −0.0134 *** | −0.8926 *** |
| | (−7.64) | (−6.23) | (−5.54) | (−5.88) |
| Top1 | −0.0027 | −0.0820 | −0.0030 | −0.0728 |
| | (−1.10) | (−0.45) | (−1.09) | (−0.37) |
| Grow | 0.0003 | 0.0148 | 0.0012 ** | 0.0800* |
| | (0.51) | (0.41) | (2.12) | (1.96) |
| _cons | 0.0169 * | −11.9868 *** | 0.0183 * | −12.4081 *** |
| | (1.86) | (−18.64) | (1.81) | (−17.25) |
| Ind/Year | YES | YES | YES | YES |
| N | 8236 | 8236 | 6094 | 6094 |
| adj. R$^2$ | 0.290 | 0.283 | 0.291 | 0.289 |

Notes: The *, **, and ***, respectively, denote the significance on the basis of two-tailed *t*-tests at or below the 10%, 5%, and 1% levels.

**Table 5.** Unabsorbed slack resources and enterprise innovation: the moderating effect of managerial ability.

| Variables | (1) Rd_Assets | (2) Patent | (3) F.Rd_Assets | (4) F.Patent |
|---|---|---|---|---|
| Slack | −0.0013 * | −0.0745 | −0.0013 * | −0.0290 |
| | (−1.65) | (−1.56) | (−1.70) | (−0.50) |
| Slack$^2$ | −0.0001 ** | −0.0093 *** | −0.0001 | −0.0105 *** |
| | (−2.33) | (−3.31) | (−1.53) | (−3.25) |
| Slack × MA_Score | 0.0004 *** | 0.0239 *** | 0.0003 *** | 0.0185 *** |
| | (4.33) | (4.96) | (4.16) | (2.98) |
| Size | 0.0001 | 0.5639 *** | 0.0000 | 0.5847 *** |
| | (0.25) | (17.86) | (0.00) | (16.33) |
| Lev | 0.0016 | −0.2316 | 0.0016 | −0.1506 |
| | (0.62) | (−1.25) | (0.55) | (−0.71) |
| Roa | 0.0339 *** | −0.3140 | 0.0449 *** | 0.3389 |
| | (6.33) | (−1.05) | (6.24) | (0.81) |
| Age | −0.0013 * | −0.0295 | −0.0010 | −0.0368 |
| | (−1.96) | (−0.63) | (−1.40) | (−0.71) |
| Board | −0.0012 | 0.2281 * | −0.0005 | 0.2478 * |
| | (−0.70) | (1.95) | (−0.26) | (1.90) |
| Bm | −0.0162 *** | −0.8134 *** | −0.0135 *** | −0.8971 *** |
| | (−7.64) | (−6.48) | (−5.64) | (−5.97) |
| Top1 | −0.0030 | −0.1114 | −0.0031 | −0.1042 |
| | (−1.21) | (−0.63) | (−1.15) | (−0.53) |
| Grow | −0.0012 ** | −0.0176 | −0.0005 | 0.0358 |
| | (−2.41) | (−0.50) | (−1.03) | (0.89) |
| _cons | 0.0178 ** | −12.0811 *** | 0.0196 ** | −12.4589 *** |
| | (1.99) | (−19.07) | (1.96) | (−17.64) |
| Controls | YES | YES | YES | YES |
| Ind/Year | YES | YES | YES | YES |
| N | 8538 | 8538 | 6350 | 6350 |
| adj. R$^2$ | 0.282 | 0.288 | 0.281 | 0.291 |

Notes: The *, **, and ***, respectively, denote the significance on the basis of two-tailed *t*-tests at or below the 10%, 5%, and 1% levels.

### 4.4. Robustness Tests

4.4.1. Replace Explained Variables

Herein, R&D investment is the natural logarithm of 1 plus the enterprise's R&D investment, and innovation output is the natural logarithm of 1 plus the patent applications number. The regression results are shown in Table A3. It can be found that the main findings above remain unchanged.

4.4.2. Replace Explanatory Variables

Firstly, following Cleary (1999) [49], this paper re-measures unabsorbed slack resources. The specific calculation formula is as follows: unabsorbed slack resources = [monetary capital + tradable financial assets + 0.7 * (net notes receivable + net receivables) + 0.5 * net

inventory—short-term borrowing]/average total assets – the index in the same year in the same industry. The new index of unabsorbed slack resources is used for the re-regression of Model (2), and the results are shown in columns (1)–(4) of Table A4. Secondly, the monthly index of China's economic policy uncertainty index provided by Baker (2016) is summed up to the annual level, and then the natural logarithm is added by 1 to obtain the economic policy uncertainty index, which is treated as a substitute for the enterprise's external environment uncertainty. The results are shown in columns (5) and (6) of Table A4. Finally, the ranking value of managerial ability is used as a substitute for managerial ability, and the results are shown in columns (7) and (8) of Table A4.

4.4.3. Transformation Estimation Method: Tobit Regression with Restricted Variables

Considering that the explained variables R&D investment and innovation output contain some observed values with a positive probability of 0, this paper employs the *Tobit* model to re-test the main hypothesis, with the regression results shown in Table A5. It can be found that the above conclusions remain unchanged after the transformation estimation method.

## 5. Further Discussion: Mechanism Analysis

The results discussed in the previous section indicate that there is an inverted U-shaped relationship between unabsorbed slack resources and enterprise innovation. In this section, we address the mechanism issue. Referring to Baron and Kenny (1986) [50], we construct the following mediating effect models:

$$M_{i,t} = \lambda_0 + \lambda_1 Slack_{i,t} + \sum Control_{i,t} + \varepsilon_{i,t} \tag{5}$$

$$RD_{i,t}/Patent_{i,t} = \delta_0 + \delta_1 Slack_{i,t} + \delta_2 M_{i,t} + \sum Control_{i,t} + \varepsilon_{i,t} \tag{6}$$

Within them, *M* is the mediating variable, which includes the financing constraints and large shareholders' tunneling. If $\lambda_1$ in Model (5) is significant, it indicates that unabsorbed slack resources have an impact on the mediator variable. In Model (6), the mediating variable *M* and unabsorbed slack resources Slack simultaneously perform regression on the explained variable of enterprise innovation. If the coefficient $\delta_2$ of the mediating variable is significant, then it indicates that the indirect effect does exist and requires future observation of the coefficient of $\delta_1$. If $\delta_1$ is significant, then the direct effect is significant, and the mediation is partial. Otherwise, the mediation is complete. In addition, to avoid the omission of mediating variables, the Bootstrap method is also used to estimate the interval of $\lambda_1 \times \delta_2$. If the estimated significant 95% confidence interval of $\lambda_1 \times \delta_2$ does not include zero, then the indirect effect is significant and the mediating effect is valid.

### 5.1. The Mediating Effect of Financing Constraints

The hypothesis we proposed in the second part states that the increase of unabsorbed slack resources will ease the financing constraints of enterprises, which directly increase the available innovation resources of enterprises. Consequently, there will be a rising stage with the increase in enterprise innovation and unabsorbed slack resources. In order to verify the above deduction, financing constraints are regarded as a mediating variable on Models (5) and (6). To ensure the reliability of research conclusions, the KZ index [51] and SA index [52] are used to measure financing constraints. First of all, in columns (1) and (2) in Table A6, the coefficient of *Slack* is significantly negative at the 1% level, indicating that unabsorbed slack resources do alleviate the financing constraints of enterprises. As can be seen from columns (3) to (6) in Table A6, from the perspective of R&D investment, although Model (6) failed the test when the SA index is used to measure financing constraints, it passed the mediating effect existence test when the KZ index was used. In terms of innovation output, the SA index passed the test, which basically verified the existence of

the mediating path of financing constraints. The results reported above provide strong evidence that unabsorbed slack resources can ease financing constraints and thus foster innovation activities, which is consistent with Daniele Amore et al. (2013) [20], Tan and Peng (2003) [21], and Marlin and W. Geiger (2015) [42].

Furthermore, 5000 random samples were collected using the bootstrap method, and the bootstrap interval estimation results in Table A8 are in line with the conclusions obtained by the stepwise regression method.

### 5.2. The Mediating Effect of Large Shareholder Tunneling

The hypothesis in Section 2 indicates that unabsorbed slack resources will restrain innovation activities when they exceed a certain level, mainly because a large number of slack resources with high flexibility may lead to large shareholders' tunneling behavior. To verify this deduction, the tunneling of large shareholders is used as a mediating variable for step-by-step regression. According to Jiang (2010) [53], this paper measures tunneling degree from the perspective of capital occupation by large shareholders. We adopt two methods to measure the large shareholders' tunneling behavior, which are as follows: Tun1 = Other receivables/Revenue, Tun2 = Other receivables/Total assets. As can be seen from columns (1) and (2) of Table A7, the coefficient of *Slack* is significantly positive at 1%, regardless of different measurements, indicating that unabsorbed slack resources intensify the tunneling behavior of large shareholders. This finding supports the principal agent theory, and it also confirms that excessive slack does result in "inefficiencies" in the enterprise [54]. From columns (3) to (6), except for the insignificant coefficient of Tun2, the coefficients are all significantly negative, and the coefficient sign of $\lambda_1 \times \delta_2$ is also negative, which is the same as $\delta_1$, indicating that the mediating effect is valid. This means that with unabsorbed slack resources appropriated by major shareholders, enterprises naturally invest fewer resources into innovation and generate lower output, which is in alignment with the findings of Chen et al. (2015) [32]. In the Bootstrap test of Table A8, 0 is not included in the interval of indirect effects, which also verifies that the mediating path of tunneling by large shareholders does exist.

## 6. Conclusions and Policy Implications

Innovation is the primary driving force for development [1], and how enterprises make resource deployments to improve technological innovation is becoming a hot topic. Based on all A-share listed companies in Shanghai and the Shenzhen Stock Market from 2011 to 2018, this paper explores the impact of unabsorbed slack resources on enterprise innovation. We discovered that there is an inverted U-shaped relationship between unabsorbed slack resources and enterprise innovation, and this inverted U-shaped relationship diminishes with environmental uncertainty and managerial ability. Furthermore, the mechanism tests reveal that increasing unabsorbed slack resources increases accessible innovation resources by alleviating financing constraints, which is favorable to technological innovation activities. However, when they exceed a certain level, they may induce large shareholders' tunneling behavior and thereby restrain the R&D investment and innovation output of enterprises.

### 6.1. Theoretical Contributions

Our findings thereby enrich the literature on unabsorbed slack resources and technological innovation in the following two ways. Firstly, by providing empirical evidence of an inverted U-shaped relationship between unabsorbed slack resources and enterprise innovation, our research makes contributions to the controversy about the consequences of slack resources. Prior study generally concentrating exclusively on slack resources assumes that they are homogeneous and ignores any differences in the flexibility of their nature. Our finding reveals that it is the unabsorbed slack resources, which can be used flexibly, that have an effect on the innovation activities. In addition, we find that slack resources are not uniformly good or bad for enterprise innovation but come to a peak to optimize innovation performance. As a consequence, this study provides a plausible explanation

for the findings of previous inconsistent studies. Secondly, we find that the effect that unabsorbed slack resources have on enterprise innovation changes with the internal and external environment, i.e., managerial ability and environmental uncertainty. The findings broaden the boundaries of the existing literature on the factors influencing technological innovation and provide evidence for social learning theory and agency theory.

### 6.2. Managerial Implications

The implications for enterprises are threefold. (1) Enterprise should optimize their internal resource structure. On the one hand, enterprises should maintain appropriate unabsorbed slack resources to cushion the turbulent environment while giving full play to R&D advantages, effectively promoting independent innovation and ultimately improving enterprise performance. On the other hand, on the premise of maintaining the absorbed slack resources required for enterprise survival and development, budget and management control should be strengthened to effectively reduce excessive absorbed slack resources to minimize their negative impact on the enterprise. (2) Enterprise should improve the adaption to the dynamic environment. Faced with a highly uncertain market environment, enterprises should rationally choose their strategic positioning, constantly pay attention to the market environment and the characteristics of their own resource endowment, accurately grasp their strategic positioning, and promote a high level of match between their resource advantages and the external market environment. Management should keep an eye on market, technology, policy, and other environmental changes; improve operational decision-making capabilities; and resist the negative impact of external environmental changes on corporate innovation. (3) Enterprise should strengthen continuous supervision of managers. The findings of this paper support the "management risk aversion hypothesis". When an enterprise has more unabsorbed slack resources, managers will tunnel the enterprise out of self-interest. As a consequence, this paper recommends that enterprises should continuously improve their internal supervision mechanisms and supervise their managers. In particular, when an enterprise possesses a large amount of unabsorbed slack resources, it is necessary to suitably restrict management discretion to prevent powerful managers from damaging enterprises' interests.

### 6.3. Limitations and Future Research

The limitations of this study provide an avenue for future research investigations. First, the quick ratio was adopted to measure the unabsorbed slack resources. Although it can reflect the actual available material resources to a certain extent, enterprises' slack resources also include non-financial resources [55]. Therefore, other appropriate measurements need to be explored in future research. Second, due to the availability of data, we chose all A-share listed companies in Shanghai and the Shenzhen Stock Market as research samples. Since China's reform and opening up, however, 65% of China's invention patents, more than 75% of its technological innovation, and 80% of its new products are completed by small- and medium-sized enterprises (SMEs). Consequently, it was essential to choose SMEs as the research sample to verify the impact of unabsorbed slack resources on enterprise innovation. Third, this paper reveals that there is a negative moderating effect between environmental uncertainty and managerial ability on unabsorbed slack resources and enterprise innovation, and exploring the mechanism that can effectively alleviate this negative moderating effect is a topic worthy of future research.

**Author Contributions:** Conceptualization, Z.S.; Data curation, Y.Z. and M.S.; Methodology, Y.Z.; Software, Y.Z. and M.S.; Supervision, Z.S.; Validation, M.S.; Writing—original draft, Y.Z. and M.S.; Writing—review & editing, Y.Z. and Z.S. All authors have read and agreed to the published version of the manuscript.

**Funding:** This research was funded by National Social Science Fund Later Funding Project of China (Grant No. 21FGLB017), the Ministry of Education Humanities and Social Sciences Research Planning

Fund (Grant No. 18YJA790070), and the Project of Postgraduate Research & Practice Innovation Program of Jiangsu Province (Grant No. KYCX21_2089).

**Institutional Review Board Statement:** Not applicable.

**Informed Consent Statement:** Not applicable.

**Data Availability Statement:** The data used to support the findings of this study are available from the corresponding author upon request.

**Conflicts of Interest:** The authors declare no conflict of interest.

## Appendix A

**Table A1.** Variable definitions.

| Variable Type | Variable Name | Variable Symbol | Variable Definition |
|---|---|---|---|
| Explained variables | R&D investment | Rd_assets | R&D investment/Total assets at the end of the period |
| | Innovation output | Patent | LN (Number of invention patents granted this year + 1) |
| Explanatory variables | Unabsorbed slack resources | Slack | (Current assets inventory)/Current liabilities |
| Moderators | Environmental uncertainty | EU | Estimated by Model (1) and adjusted by industry |
| | Managerial ability | MA_Score | Refer to Demerjian et al. (2012), using DEA phased calculations |
| Control variables | Enterprise size | Size | The natural logarithm of the book value of total assets at the end of the year |
| | Leverage | Lev | Total liabilities/total assets |
| | Return on total assets | ROA | Net profit/average balance of total assets |
| | Enterprise age | Age | LN (Current year minus listing year plus one) |
| | Independent director independence | Board | Number of independent directors/Total number of board of directors |
| | Book-to-market ratio | BM | Total assets/Total market value |
| | Equity concentration | Top1 | Number of shares held by the largest shareholder/Total share capital |
| | Enterprise growth | Grow | (Revenue of the current year minus revenue of the previous year)/Revenue of the previous year |
| | Industry | Ind | Dummy variable |
| | Year | Year | Dummy variable |

Source: The authors.

**Table A2.** Correlation analysis of main variables.

| Variables | Rd_Assets | Patent | Slack | EU | IC |
|---|---|---|---|---|---|
| Rd_assets | 1.000 | | | | |
| Patent | 0.262 *** | 1.000 | | | |
| Slack | 0.161 *** | −0.095 *** | 1.000 | | |
| EU | −0.125 *** | −0.069 *** | −0.006 | 1.000 | |
| IC | 0.070 *** | 0.065 *** | 0.072 *** | −0.116 *** | 1.000 |

Notes: The ***, respectively, denote the significance on the basis of two-tailed *t*-tests at or below the 1% levels.

## Appendix B

**Table A3.** Robustness test I: replacing dependent variables.

| Variables | (1) | (2) | (3) | (4) | (5) | (6) | (7) | (8) |
|---|---|---|---|---|---|---|---|---|
| | RD | Patent_New | RD | Patent_New | RD | Patent_New | RD | Patent_New |
| Slack | −0.1872 *** | −0.1326 *** | −0.2062 *** | −0.1164 *** | −0.1373 *** | −0.1087 *** | −0.1835 *** | −0.1337 *** |
| | (−7.55) | (−6.27) | (−7.72) | (−4.71) | (−5.22) | (−4.88) | (−7.37) | (−6.32) |
| Slack$^2$ | 0.0071 *** | 0.0044 ** | 0.0096 *** | 0.0031 | 0.0083 *** | 0.0050 ** | 0.0069 *** | 0.0045 ** |
| | (2.94) | (2.22) | (3.98) | (1.33) | (3.57) | (2.49) | (2.91) | (2.24) |
| Slack × EU | | | | | −0.0481 *** | −0.0231 *** | | |
| | | | | | (−7.16) | (−3.90) | | |
| Slack × MA_Score | | | | | | | 0.1879 *** | −0.0601 * |
| | | | | | | | (3.97) | (−1.69) |
| _cons | 15.9326 *** | 0.5329 *** | 16.3858 *** | 0.6960 *** | 15.9308 *** | 0.5321 *** | 15.9259 *** | 0.5350 *** |
| | (102.48) | (4.20) | (96.53) | (5.00) | (102.00) | (4.17) | (102.53) | (4.22) |
| Controls | YES | YES | YES | YES | YES | YES | YES | YES |
| Ind/Year | YES | YES | YES | YES | YES | YES | YES | YES |
| N | 8236 | 8236 | 6094 | 6094 | 8236 | 8236 | 8236 | 8236 |
| adj. R$^2$ | 0.165 | 0.140 | 0.159 | 0.141 | 0.171 | 0.142 | 0.167 | 0.141 |

Notes: The *, **, and ***, respectively, denote the significance on the basis of two-tailed *t*-tests at or below the 10%, 5%, and 1% levels.

**Table A4.** Robustness test II: replacing independent variables.

| Variables | (1) | (2) | (3) | (4) | (5) | (6) | (7) | (8) |
|---|---|---|---|---|---|---|---|---|
| | Rd_Assets | Patent | F.Rd_Assets | F.Patent | Rd_Assets | Patent | Rd_Assets | Patent |
| Slacknew | 0.0153 *** | 0.9533 *** | 0.0137 *** | 0.9042 *** | 0.0157 *** | 0.9839 *** | 0.0160 *** | 1.0041 *** |
| | (7.23) | (6.98) | (5.82) | (5.98) | (7.33) | (7.15) | (7.51) | (7.35) |
| Slacknew$^2$ | −0.0223 *** | −1.6824 *** | −0.0188 ** | −1.8182 *** | −0.0204 *** | −1.5248 *** | −0.0185 *** | −1.3995 *** |
| | (−3.34) | (−3.25) | (−2.54) | (−3.17) | (−3.03) | (−2.92) | (−2.76) | (−2.68) |
| Slacknew × EPU_Baker | | | | | −0.0001 ** | −0.0061*** | | |
| | | | | | (−2.16) | (−3.29) | | |
| Slacknew × MA_Rank | | | | | | | −0.0002 *** | −0.0163 *** |
| | | | | | | | (−3.84) | (−4.50) |
| _cons | 0.0180 ** | −11.9502 *** | 0.0199 ** | −12.3066 *** | 0.0189 ** | −11.8704 *** | 0.0185 ** | −11.9083 *** |
| | (2.01) | (−19.22) | (2.00) | (−17.64) | (2.12) | (−19.10) | (2.07) | (−19.22) |
| Controls | YES | YES | YES | YES | YES | YES | YES | YES |
| Ind/Year | YES | YES | YES | YES | YES | YES | YES | YES |
| N | 8236 | 8236 | 6094 | 6094 | 8236 | 8236 | 8236 | 8236 |
| adj. R$^2$ | 0.298 | 0.295 | 0.294 | 0.298 | 0.300 | 0.297 | 0.302 | 0.299 |

Notes: The **, and ***, respectively, denote the significance on the basis of two-tailed *t*-tests at or below the 5%, and 1% levels.

**Table A5.** Robustness test III: Tobit regression with restricted variables.

| Variables | (1) | (2) | (3) | (4) | (5) | (6) | (7) | (8) |
|---|---|---|---|---|---|---|---|---|
| | Rd_Assets | Patent | F.Rd_Assets | F.Patent | Rd_Assets | Patent | Rd_Assets | Patent |
| Slack | 0.0012 *** | 0.0789 ** | 0.0010 ** | 0.1023 *** | 0.0013 *** | 0.0767 ** | 0.0020 *** | 0.1077 *** |
| | (3.59) | (2.37) | (2.56) | (2.70) | (3.69) | (2.32) | (5.81) | (3.16) |
| Slack$^2$ | −0.0001 *** | −0.0124 *** | −0.0001 *** | −0.0147 *** | −0.0001 *** | −0.0120 *** | −0.0001 *** | −0.0119 *** |
| | (−4.36) | (−4.41) | (−2.93) | (−4.66) | (−4.46) | (−4.29) | (−3.81) | (−4.22) |
| Slack × EU | | | | | −0.0007 *** | −0.0261 *** | | |
| | | | | | (−10.80) | (−3.86) | | |
| Slack × MA_Score | | | | | | | −0.0031 *** | −0.3120 *** |
| | | | | | | | (−6.69) | (−6.94) |
| _cons | 0.0173 *** | −15.4150 *** | 0.0191 *** | −15.6130 *** | 0.0169 *** | −15.4205 *** | 0.0159 *** | −15.5347 *** |
| | (3.79) | (−34.95) | (3.51) | (−30.25) | (3.73) | (−34.99) | (3.49) | (−35.31) |
| Controls | YES | YES | YES | YES | YES | YES | YES | YES |
| Ind/Year | YES | YES | YES | YES | YES | YES | YES | YES |
| N | 8236 | 8236 | 6094 | 6094 | 8236 | 8236 | 8236 | 8236 |

Notes: The **, and ***, respectively, denote the significance on the basis of two-tailed *t*-tests at or below the 5%, and 1% levels.

## Appendix C

**Table A6.** Mechanism I: Financing constraints (Resource effect).

| Variables | (1) | (2) | (3) | (4) | (5) | (6) |
|---|---|---|---|---|---|---|
| | SA | KZ | Rd_Assets | Patent | Rd_Assets | Patent |
| Slack | −0.0048 *** | −0.1236 *** | −0.0002 | −0.0301 ** | −0.0002 | −0.0297 ** |
| | (−3.42) | (−6.01) | (−0.71) | (−2.24) | (−0.94) | (−2.19) |
| SA | | | −0.0019 | −0.6255 *** | | |
| | | | (−0.61) | (−2.62) | | |
| KZ | | | | | −0.0005 ** | −0.0212 |
| | | | | | (−2.39) | (−1.56) |
| _cons | 3.3918 *** | 4.7044 *** | 0.0259 * | −9.7960 *** | 0.0221 ** | −11.8177 *** |
| | (35.12) | (8.48) | (1.80) | (−11.30) | (2.45) | (−18.40) |
| Controls | YES | YES | YES | YES | YES | YES |
| Ind/Year | YES | YES | YES | YES | YES | YES |
| N | 8538 | 8538 | 8538 | 8538 | 8538 | 8538 |
| adj. R$^2$ | 0.854 | 0.583 | 0.279 | 0.287 | 0.281 | 0.285 |

Notes: The *, **, and ***, respectively, denote the significance on the basis of two-tailed *t*-tests at or below the 10%, 5%, and 1% levels.

**Table A7.** Mechanism II: Tunneling by large shareholders (Agency cost).

| Variables | (1) | (2) | (3) | (4) | (5) | (6) |
|---|---|---|---|---|---|---|
| | Tun1 | Tun2 | Rd_Assets | Patent | Rd_Assets | Patent |
| Slack | 0.0032 *** | 0.0010 *** | −0.0001 | −0.0250 * | −0.0001 | −0.0277 ** |
| | (2.76) | (3.36) | (−0.28) | (−1.86) | (−0.53) | (−2.05) |
| Tun1 | | | −0.0279 *** | −0.6386 *** | | |
| | | | (−8.60) | (−2.65) | | |
| Tun2 | | | | | −0.0307 *** | 0.6324 |
| | | | | | (−2.71) | (0.72) |
| _cons | 0.0821 *** | 0.0275 *** | 0.0219 ** | −11.8627 *** | 0.0204 ** | −11.9326 *** |
| | (2.70) | (2.67) | (2.48) | (−18.71) | (2.29) | (−18.77) |
| Controls | YES | YES | YES | YES | YES | YES |
| Ind/Year | YES | YES | YES | YES | YES | YES |
| N | 8533 | 8533 | 8533 | 8533 | 8533 | 8533 |
| adj. $R^2$ | 0.108 | 0.122 | 0.289 | 0.285 | 0.280 | 0.285 |

Notes: The *, **, and ***, respectively, denote the significance on the basis of two-tailed $t$-tests at or below the 10%, 5%, and 1% levels.

## Appendix D

**Table A8.** Bootstrap analysis for mediation effect ($N$ = 5000).

| Mediating Path | The Indirect Effect | | | The Direct Effect | | |
|---|---|---|---|---|---|---|
| | Effect | 95% Confidence Interval | | Effect | 95% Confidence Interval | |
| | | BootLLCI | BootULCI | | BootLLCI | BootULCI |
| Slack→SA→Rd_assets | 0.000000 | −0.000010 | 0.000030 | 0.000327 | 0.000038 | 0.000616 |
| Slack→SA→Patent | 0.002562 | 0.000763 | 0.004361 | −0.013846 | −0.029808 | 0.002117 |
| Slack→KZ→Rd_assets | 0.000042 | 0.000000 | 0.000080 | 0.000294 | 0.000000 | 0.000577 |
| Slack→KZ→Patent | 0.000274 | −0.001963 | 0.002511 | −0.011558 | −0.027616 | 0.004501 |
| Slack→Tun1→Rd_assets | −0.000101 | −0.000139 | −0.000063 | 0.000445 | 0.000154 | 0.000736 |
| Slack→Tun1→Patent | −0.004803 | −0.007092 | −0.002514 | −0.006155 | −0.022430 | 0.010119 |
| Slack→Tun2→Rd_assets | −0.000041 | −0.000064 | −0.000017 | 0.000385 | 0.000092 | 0.000678 |
| Slack→Tun2→Patent | −0.002105 | −0.003816 | −0.000395 | −0.008853 | −0.025666 | 0.007960 |

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
