# Peer review of "Unabsorbed Slack Resources and Enterprise Innovation: The Moderating Effect of Environmental Uncertainty and Managerial Ability"

_sustainability, doi:10.3390/su14073782_

Round 1

Reviewer 1 Report

This manuscript explores the impact of unabsorbed slack resources on enterprise innovation from two dimensions of R&D investment and innovation output with the data of all A-share listed companies in Shanghai and Shenzhen Stock Market from 2011 to 2018. 

This manuscript enhances the theoretical basis of the smoothing effect of unabsorbed slack resources on enterprise innovation. And, from environmental uncertainty and managerial ability, this paper de scribes the moderating role of internal and external factors on the relationship between unabsorbed slack resources and enterprise innovation, reveals the mechanisms by which enterprises choose to accumulate or consume unabsorbed slack resources in different contexts.

However, the Introduction, Literature review and research hypothesis, Sample selection and data sources, Further discussion: Mechanism analysis sections of this manuscript are weak. The specific comments are as follows: 

1) In the Introduction section, please clearly indicate the purpose and the scientific question of this manuscript â€‹to be addressed.

2) Please explain the meaning of figure 1 in section 2 instead of just listing it.

3) Although the views of many researchers are listed in section 2, they are insufficiently critical. In addition, some important literature in the R&D field were left out.

e.g.,

Barker, V. L., & Mueller, G. C. (2002). CEO characteristics and firm R&D spending. Management Science, 48(6), 782-801.

Becker, W., & Dietz, J. (2004). R&D cooperation and innovation activities of firms—evidence for the German manufacturing industry. Research policy, 33(2), 209-223.

Cohen, W. M., Nelson, R. R., & Walsh, J. P. (2002). Links and impacts: the influence of public research on industrial R&D. Management science, 48(1), 1-23.

De Vita, G., Li, C., & Luo, Y. (2021). The inward FDI-Energy intensity nexus in OECD countries: A sectoral R&D threshold analysis. Journal of Environmental Management, 287, 112290.

Du, J. , Zhang, J. , & Li, X. . (2020). What is the mechanism of resource dependence and high-quality economic development? an empirical test from china. Sustainability, 12(19), 8144.

Gassmann, O., & Von Zedtwitz, M. (1999). New concepts and trends in international R&D organization. Research policy, 28(2-3), 231-250.

Guellec, D., & Van Pottelsberghe De La Potterie, B. (2003). The impact of public R&D expenditure on business R&D. Economics of innovation and new technology, 12(3), 225-243.

Hall, B. H., & Mairesse, J. (1995). Exploring the relationship between R&D and productivity in French manufacturing firms. Journal of econometrics, 65(1), 263-293.

Kuemmerle, W. (1997). Building effective R&D capabilities abroad. Harvard business review, 75, 61-72.

Leung, T. Y., & Sharma, P. (2021). Differences in the impact of R&D intensity and R&D internationalization on firm performance–Mediating role of innovation performance. Journal of Business Research, 131, 81-91.

Li, X. , & Long, H. . (2020). Research focus, frontier and knowledge base of green technology in china: metrological research based on mapping knowledge domains. Polish Journal of Environmental Studies, 29(5), 3003–3011.

Safi, A., Chen, Y., Wahab, S., Zheng, L., & Rjoub, H. (2021). Does environmental taxes achieve the carbon neutrality target of G7 economies? Evaluating the importance of environmental R&D. Journal of Environmental Management, 293, 112908. 

etc.

4) In the section 3.1. Sample selection and data sources, please explain the objective reasons for selecting the data. Also, please add references to the data sources for this manuscript.

5) In the Further discussion: Mechanism analysis section, in order to improve the academic contribution of this manuscript, please compare the similarities and differences between the results of this study and similar studies extensively and deeply.

All in all, this manuscript is interesting. The authors are invited to carefully revise this manuscript in accordance with the above suggestions. I sincerely look forward to receiving the revised version. 

Reviewer 2 Report

The paper is interesting, it deals with an important topic of unabsorbed slack resources and enterprise innovation, and it is my pleasure to review it.

The paper has merits, is very detailed, well organized, and uses a solid scientific and logical tool. Literature, methodology and approaches are interesting, systematic and comprehensive.

However, I would have some considerations and suggestions for improving the quality of the article.

The titles of the figures should be more explicit, to better suggest their content / meaning (see Fig. 1).

The Conclusions, although well-constructed and comprehensive, advance some deductions that have been partially or indirectly analysed in the paper. See risk aversion, preventing managers with too strong personal ability to harm enterprise etc. They are of interest for the practical, applicative value of the research, but they seem to be quite artificially introduced in this section.

This can also be a consequence of an imbalance between the sophisticated analytical construction, loaded with calculations and abstractions, and, respectively, the need for a pragmatic final product, operationalizable in the economic and managerial activity of these companies.

The paper does not prove (probably it does not even propose it) any reference or consistent connection with sustainability (except for a single indirect reference, in the literature review part), regardless of its perspective (environmental, economic, social, etc.).

Minor formal issues

- Markers as â‘ , â‘¡ is quite unusual, and sometimes confusing

- Double counting of Final References

Thank you for the opportunity to review this article and good luck!

Reviewer 3 Report

Text

The body of text extensively uses the phrasing “On the other hand” (i.e., 172~174), please replace some of the wording.

Methodology

Although Research Design is well explained and supported by literature, the used methodology is not so. Describe the methodology used in the research, supported by literature.

Conclusions

Segregate and describe separately the theoretical and practical contributions

Round 2

Reviewer 1 Report

The authors carefully revised their manuscript and recommended that it be accepted.